# A Possible Natural and Inexpensive Substitute for Lapis Lazuli in the Frederick II Era: The Finding of Haüyne in Blue Lead-Tin Glazed Pottery from Melfi Castle (Italy)

**DOI:** 10.3390/molecules28041546

**Published:** 2023-02-06

**Authors:** Annarosa Mangone, Maria Cristina Caggiani, Tiziana Forleo, Lorena Carla Giannossa, Pasquale Acquafredda

**Affiliations:** 1Department of Chemistry, University of Bari “Aldo Moro”, 70125 Bari, Italy; 2Interdepartmental Centre “Research Laboratory for the Diagnostics of Cultural Heritage”, University of Bari “Aldo Moro”, 70125 Bari, Italy; 3Department of Biological, Geological and Environmental Sciences, University of Catania, 95129 Catania, Italy; 4Department of Earth and Geoenvironmental Sciences, University of Bari “Aldo Moro”, 70125 Bari, Italy

**Keywords:** blue, lapis lazuli, haüyne, medieval blue tin-lead glazed ceramic, Frederick II

## Abstract

The blue color of glass and ceramic glazes produced in Apulia and Basilicata (Southern Italy) between the 13th and 14th centuries and connected to the Norman-Swabian Emperor Frederick II, has been, for a long time, under archaeometric investigation. On the one hand, it has usually been associated with lapis lazuli, due to the finding of the polysulphide blue chromophores typical of lazurite. Moreover, the observation that the mineral haüyne, which belongs to the sodalite group as well as lazurite, can be blue and/or can gain a blue color after heating, due to the same chromophores, has caused this automatic attribution to be questioned, and also considering that the mineral is characteristic of the rock haüynophyre of Melfi (Potenza, Southern Italy), a location of interest for glass and pottery findings. In this paper, for the first time, several haüyne crystals were found in the blue glaze of a ceramic dish found at Melfi Castle, leading to the hypothesis that, in this case, the local haüyne-bearing source could have been used as the coloring raw material. The discovery was possible thanks to SEM-EDS and Raman analyses that, respectively, highlighted the typical numerous presence of very fine sulphur-based inclusions in the crystals and the characteristic Raman signal of blue haüyne. This study was also focused on the composition of the crystals inclusions, aided by SEM-EDS and Raman maps, since the original very fine pyrrhotite was transformed into Cu and Pb phases (copper sulphates, copper sulphides, and lead oxide) due to reactions with cations that had mobilized from the glaze, while the migration of Si from the glass allowed the transformation of the rim of the haüyne, a silica-undersaturated mineral, into a corona of small euhedral and neomorphic Pb-rich feldspars, a silica-saturated phase.

## 1. Introduction

Lazurite [(Na,Ca)_8_Al_6_Si_6_O_24_(SO_4_,S,Cl)_2_] belongs to the mineral group of sodalite which mainly includes three other minerals: sodalite *s.s.* (Na_8_Al_6_Si_6_O_24_Cl_2_), nosean (Na_8_Al_6_Si_6_O_24_SO_4_), and haüyne [Na_6_Ca_2_Al_6_Si_6_O_24_(SO_4_)_2_]. In nature, lazurite, sodalite *s.s.*, and sometimes haüyne can be found with a blue color, due to the presence of $S_{3}^{-}$ groups [1,2,3,4,5]. The peculiarity of lazurite is not only related to its intense blue hue but also to the quite unchangeable characteristic of the color over time, essentially due to the idiochromatic character of the coloration.

Sodalite-bearing rocks are not widespread; in the Italian peninsula, they are present in the Vulsini, Vico, Alban hills, Ventotene-Santo Stefano, Vesuvius, and Vulture outcrops ([6,7,8] and references therein). 

Haüyne crystals present in the Mt. Vulture volcano area (mainly in phonolite and haüynophyre rocks) are white, black, and blue in color [2], and are almost always characterized by the presence of extremely fine inclusions of pyrrhotite (down to 50 nm in size), disposed along the cleavage of the mineral, especially near its rim, in coronitic texture [2], so much so, as to macroscopically give a pseudomorphosis of pyrrhotite on haüyne (the black crystals).

Lapis lazuli refers to a rock which is rich in lazurite. There are only a few sources of lapis lazuli in the world due to the peculiar features of geological conditions in which it can be formed (temperature and pressure of the metamorphism, oxygen and sulphur fugacity, and protolith chemistry) and the possibility of associating the raw material with man-made objects that would help to reconstruct trade routes. Rare and therefore expensive, its trade was guaranteed since antiquity via sea and land routes that, according to some experts, were significantly blocked in the late Middle Ages [9]. The use of lapis lazuli is well documented in frescoes, paintings, and illuminated manuscripts in different parts of the world [10], whereas its use in everyday objects such as glass and glazed ceramic is rarer and, for a long time, it has been ignored such as, for example, in the Ptolemaic faience, in which the first finding of lapis lazuli was relatively recent [11]. 

This study is part of a broader research topic concerning pigmenting minerals used in medieval vitreous and fictile production in Southern Italy [1,12,13,14] with a particular interest in understanding the reasons for the widespread use of lapis lazuli to color blue glass and glazed ceramics in Apulia and Basilicata between the 13th and 14th centuries. 

Archaeometric investigations have, indeed, highlighted the use of lapis lazuli in lead-tin glazed ceramic from the abandoned medieval village of Castel Fiorentino [12,15], in a bowl discovered in the Swabian-Angevin fortress of Lucera [15], in the only surviving mosaic tile belonging to the original mosaic decoration of Castel del Monte [15], and in several fragments of an atypical protomajolica found in the archaeological site of Siponto [13]. The contexts examined are all court residences linked to the Swabian sovereign: the fortress of Lucera, seat of a large Saracen colony deported there at the behest of Frederick II, in which there is evidence of the work of Islamic ceramists; Castel Fiorentino, an imperial domus traditionally identified as the emperor’s last residence; Castel del Monte, the most famous and problematic of the residences built by the Swabian sovereign; the town of Siponto, also linked to the figure of Emperor Frederick II, and one of the busiest ports for those heading to and from the Middle East. 

The widespread use of lapis lazuli as a pigment in 13th century and 14th century glazed ceramics in Apulia, far from being isolated in a few pieces, thus, seems to have been more common than previously thought and suggests the existence of a local production that imitates or is influenced by Islamic products [16,17], even if no production center has yet been identified. The influence of the Arab world on local production, during the time of Emperor Frederick II, has also been hypothesized on the basis of the artistic-technological similarity between the mosque lamps from the Mamluk period, exhibited in the Louvre in Paris, and the glass artefacts found in the Melfi Castle [1,18,19], also a residence of Emperor Frederick II, which shows how the two regions—Apulia and Basilicata—played a central role in trade with the East at the time and how Islamic art and culture influenced the area. The idea of a technological transfer between Southern Italy and the Middle East is supported by the use of lazurite chromophore in ceramic and glass decorations, and also by the presence of other technological devices typical of Middle Eastern production, such as the use of calcium phosphate as an undercoat to the glaze [1,13,20,21].

One might ask how this contamination took place and how the craftsmen learned this artistic taste, but above all, whether the artefacts were produced locally, whether the raw materials were imported, and especially from where. The fact, however, that Melfi Castle is built on and not far from outcrops of volcanic rocks (phonolite of Toppo San Paolo and haüynophyre of Melfi) that contain haüyne, which can be blue or turn blue after heating at high temperatures, has cast doubt on the true nature of the blue raw materials. Recently, indeed, on the one hand, archaeometric studies have advanced the hypothesis that the vitreous materials found in Melfi Castle were produced locally with indigenous raw materials, and that the Arab influence could be limited to technological procedures and style, but not necessarily involving raw materials supply [1,19]. On the other hand, up to now, no proof that haüyne-bearing rocks could be actually used in place of lapis lazuli to give the blue color to these types of artefacts has been found, therefore, the aim of this work was to test the blue glaze of a dish discovered within the described Melfi Castle context to try to answer this question.

## 2. Results

### 2.1. SEM-EDS

The paste of the M22 sample is characterized by a silty-sandy texture with sharp edges, quartz crystals, K-feldspars, plagioclases (mainly sodic), opaques, Fe-hydroxides, micas (mainly biotite) and, to a lesser extent, rutile, ilmenite, and rare monazite. Numerous *chamotte* are present. The sintering degree is very high. The chemical and minero-petrographic characteristics are compatible with the use of local raw materials and the manufacturing process does not appear to be particularly accurate, as suggested by the coarse grain size and by the absence of iso-orientation micas (muscovite and biotite) and pores.

On the surface, appearing strongly degraded (Figure 1 and Figure 2), a mono-stratified lead-stanniferous glaze was applied. In the black decorated areas, SEM investigations revealed the presence of Mn oxides, in the blue areas of sodalite crystals.

In addition, sodalite crystals, whose ED spectrum is typical of minerals such as lazurite or haüyne (Figure 3), show extremely fine inclusions, very bright in the BSE image (Figure 1), that are absent in lazurite while they are characteristic of haüyne.

Quantitative microanalyses of the sodalite group phases present in the lead-tin glaze of the M22 sample (Figure 1 and Figure 2) confirmed that their mean composition corresponded to the haüyne (-lazurite) field in the classification diagram of Lessing and Grout [22] for sodalite minerals (Figure 4). To compare the haüyne analyses of the M22 sample with the haüyne of the Monte Vulture area, in the same K_2_O-Na_2_O-CaO diagram, haüyne compositions were also plotted from phonolite of the Toppo San Paolo area (Vulture outcrop 40°58′47″ N–3°12′57″ E) and from haüynophyre of the Melfi area (Vulture outcrop 40°59′56″ N–3°12′14″ E), both heated at 750 °C. Moreover, the analyses of lazurites of lapis lazuli from Sar-e Sang (Badakhshan, Afghanistan) and Condoriaco (Chile) from the Earth Science Museum of the Bari University are also reported for relative estimation (Figure 4).

Quantitative SEM-EDS analyses give clear indications that the blue crystals in the lead-tin glaze of the Frederician pottery from Melfi (sample M22) belong to the sodalite group and in particular they are haüyne (Figure 4).

To further constrain the data, haüyne analyses were also compared with those present in the Monte Vulture area rocks, specifically with the crystals of phonolite (Toppo San Paolo outcrops) and haüynophyre (Melfi area) volcanics, heated at 750 °C.

Moreover, to compare haüyne heated crystals with lazurite heated crystals, microanalyses of lazurite found in Melfi (PZ) blue gilded and enameled glassware [1] were also added.

A quick look at Figure 4 shows that lazurite and haüyne lie in a circumscribed Na-rich area of the diagram; the lazurite from Melfi ceramics, that is present in a Na-rich blue enamel (Na_2_O > 13.0 wt%; data from [1]) also rich in SiO_2_ (>66.0 wt%; data from [1]), while the haüyne from sample M22 is present in a lead glass poor in SiO_2_ (<38.0 wt%) and in Na_2_O (<2.5 wt%). The different compositions of the glass in which lazurite and haüyne were heated, slightly changed the composition of lazurite (Na_2_O decreases slightly) during its heating, while for sample M22, during the firing processes, an appreciable migration of sodium from the haüyne (Na_2_O about 14 wt%) to the enamel (Na_2_O about 2.5 wt%) took place, enriching passively the mineral composition in K_2_O and CaO (Figure 4). 

The different mobilities of the chemical elements depended on the different starting concentrations in the glass and in the minerals, but also on the even slightly different structures of the minerals, and was strongly conditioned by the firing time of the artefacts.

The blue haüyne crystals present in the lead-tin glaze of the investigated pottery are not far from the compositions of haüyne from the Monte Vulture rocks, especially from those of Melfi haüynophyre, which also, naturally and at room temperature, exhibit blue haüyne crystals, whereas haüyne from the phonolite of Toppo San Paolo outcrops shows only white or black crystals. It is worth mentioning that even white or black haüyne, with heating to at least 400 °C, turns blue [2]. 

The SEM-EDS microanalytical data (Figure 5) strongly stress the more simple and indubitable information that comes simply from petrographic observations: a sodalite group blue mineral, with numerous very small opaque inclusions, is an haüyne, typical of the Monte Vulture rocks that, among Italian volcanoes, contain this mineral as the most important foid [8].

SEM investigations also highlight that the lead-tin glaze of the ceramic M22 sample is very rich in Pb and, subordinately, in Sn and Fe; the presence of small amounts (<0.3 wt%) of elements such as Ti, Mn, Co, Cu, and Zn were detected by acquiring glaze ED spectra over very long time periods (>100 min) in order to increase the counting statistics and improve the signal-to-noise ratio of the microchemical data (Figure 3). SEM-EDS X-ray maps, clearly highlight that the fine inclusions in the haüyne of the glaze of the ceramic M22 sample are composed mainly of Pb, Cu, and S; moreover, the presence of an evident potassium-rich rim around the foid crystal, should be noted (Figure 6).

### 2.2. Raman Spectroscopy

Figure 7 exhibits representative Raman spectra extracted from the mappings (Appendix A) carried out on haüyne crystals 3 and 5. The signature of haüyne, with its main bands at about 440 and 980 cm^−1^ [2] was found, even though with different intensities or associated with other signals, in the whole crystals’ areas (Figure 7a). A thinner band centerd around 990 cm^−1^, instead, together with two weaker signals at about 620 and 635 cm^−1^, could be associated to sulphate phases presence (Figure 7b), likely copper sulphates [24] whose distribution is not uniform along the haüyne crystals. Actually, a significant part of haüyne 3 seems to be dominated by another phase revealed in both crystals by the mapping of the spectral region included between 455 and 490 cm^−1^: spectra were characterized by a main strong peak at 470 cm^−1^ (Figure 7c). This can be attributed to covellite mineral (CuS) [25]. All around haüyne crystals (see Appendix A), the Raman features of K-feldspar [26], often associated to those of haüyne itself and of glass, can be observed at 156, 284, 472 and 512 cm^−1^ (Figure 7d).

Moving away from haüyne, these peaks disappear, and around 480 and 990 cm^−1^, broad bands appear that are characteristic of lead glass [27,28] in association with the principal peak of cassiterite (SnO_2_) at 633 cm^−1^ [27] (Figure 7e). During the study of haüyne 5, other Raman signals were revealed. A sharp peak at about 140 cm^−1^ is associated to a broader band of variable intensity at ca. 280 cm^−1^ (Figure 7f, g), which in some cases appears to be isolated from the previous signal (Figure 7h). If 140 cm^−1^ peak is assigned to lead oxide (PbO), the co-presence of the band at 280 cm^−1^ could be ascribed to lead oxide alteration in the form of plattnerite (PbO_2_) under the laser beam [29]; also, the formation of lead-tin yellow, that gives a Raman band between 135 and 140 cm^−1^ [18,30], cannot be excluded. Furthermore, a possible attribution of ~140 cm^−1^ signal to anatase must be taken into account, even more so considering that this mineral was already found in haüyne-bearing rocks [1].

Furthermore, Raman analyses randomly carried out on other blue haüyne crystals, also highlighted, apart from the constant signature of haüyne itself and, frequently, other spectra already extracted from the mappings, the principal signals of $S_{3}^{-}$ and $S_{2}^{-}$ blue chromophores at 542 and 574 cm^−1^ (Figure 8) [16].

## 3. Discussions

SEM-EDS and Raman characterization of the glaze of the M22 sample from Melfi, referable to the Frederician period, revealed the presence of haüyne crystals used as pigment for the blue.

SEM-EDS investigations highlight that the very fine inclusions present in the haüyne are composed mainly of Pb, Cu, and S; copper sulphides and sulphates as well as lead oxides were also revealed by means of Raman spectroscopy. In addition, the Monte Vulture rock haüyne crystals are characterized by fine inclusions of pyrrhotite (Fe_1−x_S) [2]. 

A few observations can be summarized about the elements mobility in the considered system. From the analysis of the lead-tin glaze, which is very rich in Pb and also shows small amounts of Cu, it can be inferred that the heating processes facilitate the diffusion of Pb and Cu from the tin-lead glaze into the haüyne, transforming pyrrhotite into other phases that are stable in the new physico-chemical conditions and also leading to the formation of the $S_{3}^{-}$ chromophore group. Similar reactions were described for haüyne from Monte Vulture volcanics involving the mobility of elements from surrounding minerals in the rock when it was heated at 750 °C [2]. The Na_2_O concentrations of the haüyne present in the glaze, as compared with those of the Monte Vulture phonolite and haüynophyre (Figure 4), suggest that this element, always highly mobile in silicate systems, diffuses from the haüyne to the glaze. The reason why other chemical elements present in the glaze, such as Sn and Co, were not mobilized during the heating process, leading to the transformation of pyrrhotite into Sn- and Co-rich phases should be evaluated with great caution, but a few observations can be pointed out: (i) The mobility of some elements between the glaze and the haüyne crystals, in some cases, is justified by the abundance of the element (e.g., for Pb) or by their mobility based on ionic potential [31]. (ii) It is also necessary to keep in mind that mobility depends on the system being considered and the crystal phase in which the element is fixed; normally Pb is classified as a moderately mobile element in magmatic and metamorphic systems [31], but if present in a mineral such as monazite can be considered to be almost immobile even for very long times (>100 Ma [32]). Apparently, in this case, the mobility of the chemical elements depends not only on their concentration but also on the physico-chemical conditions, and even more on the possibility of stabilizing specific phases under the new conditions of temperature, especially taking into account oxygen and sulphur fugacity [2].

Normally, the high concentration and mobility of Pb in the coating of medieval/renaissance pottery, led to the neoformation of a Pb-rich corona around relict feldspars; in the corona, small euhedral and neomorphic Pb-rich crystals developed at the rim of the alkali feldspar in contact with the Pb-rich lead-tin glaze [27,33]. Similarly, in the M22 sample, the high mobility of some elements such as Si, K, and Pb allowed the formation of a K-rich corona around haüyne; in fact, the rim of haüyne, a silica-undersaturated mineral, reacted with Si and Pb, which are relatively more abundant in the lead-tin glaze, to form more silica-saturated crystals such as small neomorphic and euhedral Pb-rich-feldspar-like crystals, whose Raman spectrum was here highlighted and already found in [27]. The concentration of K in the haüyne and in the lead-tin glaze suggests that this element moved from the glaze to the haüyne forming the K-rich corona and enriching the haüyne in K; in fact, the K_2_O of the Monte Vulture haüyne is about 1.73 wt% while that of the haüyne in the M22 sample is higher than 2.8 wt% (Figure 3 and Figure 4).

The absence of Cu in the corona is probably due to the fact that, in the framework of the neo-formed feldspar, this element is not compatible.

## 4. Materials and Methods 

### 4.1. Sample

Fragments of glazed ceramic were found, along with a group of highly refined fragments of polychrome enameled and gilded glasses, during an archaeological excavation in the landfill of the Marcangione Tower of the Melfi Castle (Potenza, Italy). For the most part, they have been attributable to the Frederician period (12th–13th century). The archaeological study of the ceramic materials is still ongoing, also, because the discovery of these fragments prompted an overall re-examination of the larger ceramic group from excavations carried out, occasionally, in the entire castle area. Preliminary investigations have, however, ascertained among the productions identified, alongside artefacts testifying to Apulian and Campanian contributions, some specimens underlining cultural links with the Islamic world [34]. Within this group of fragments, particular interest was aroused by sample M22 for the peculiar, unique, blue shade of its decorations, very similar to that of the blue decorations attributed to lapis lazuli present on the protomajolica artefacts found in other medieval sites in Puglia (Siponto, Castel Fiorentino, and Lucera).

M22 (Figure 1) is a fragment of a dish with a horizontal brim. The decoration on the brim with a slightly oblique S-shaped motif is contained within concentric lines in brown, while a phytomorphic motif is probably present in the *cavetto*, in the center. 

A fragment of a few millimetres was sampled in order to realize a stratigraphic section to be analyzed.

### 4.2. Techniques

For the SEM investigations, samples were previously sputtered with a 30 nm thick carbon film using an Edwards Auto 306 (Edwards Vacuum, West Sussex, UK) thermal evaporator, and then fixed on aluminum stubs with adhesive aluminum conductive tape.

Microanalyses and X-ray maps were performed using a SEM of LEO, model EVO50XVP (LEO EVO-50XVP, Zeiss, Cambridge, Cambridgeshire, UK) coupled with an X-max (80 mm^2^) Silicon drift Oxford detector (Oxford Instruments, High Wycombe, Buckinghamshire, UK) equipped with a Super Atmosphere Thin Window ©. SEM operative conditions were: 15 kV accelerating potential and 200 pA probe current. More detailed information on the operating conditions can be found in the Appendix A.

Raman investigations were performed employing a LabRAM HR Evolution (Horiba, Japan) spectrometer (equipped with an Ar^+^ 514 nm laser) coupled with a BH2 microscope (Olympus, Tokyo, Japan) and a Peltier-cooled charge-coupled device detector (CCD). The spectral resolution was about 1 cm^−1^ (1800 g/mm grating) and the spatial lateral resolution was less than 1 μm. The system was calibrated using the 520.7 cm^−1^ Raman band of silicon. A linear baseline was subtracted from the raw spectra using the software LabSpec6^®^ (Horiba^®^, Kyoto, Japan). 

Prior to the maps acquisition, the parameters used were adjusted after several tests to find a compromise among intensity, noise abatement, and total performance duration, and to ensure that the samples were not altered under the laser beam. 30 s measurements with three repetitions and a power of about 10 mW were chosen. The micro-mappings were acquired with a grid of a sufficient number of spots to cover the entire surface of the crystal under examination at 2 μm step.

## 5. Conclusions

From both the archaeometric and archaeological points of view, the obtained results about a specific aspect of the ceramic production in the Frederician period led to interesting outcomes. Indeed, significant attention has been paid, in recent years, to the identification of the chromophore that gives the blue color to the glaze of proto-majolica pottery [15,35,36,37] and to the glass [18,19] found in the castles and domus of Emporer Frederick II in Basilicata and northern Apulia. In the materials studied so far, the presence of polysulfide chromophores, generally associated with lazurite and thus with lapis lazuli, has been revealed, in addition to or as an alternative to the use of cobalt, leading to the conclusion that a material often thought to be very expensive and difficult to obtain, and thus, rarely used, was actually more widespread than envisaged [10].

One very important finding was the discovery, in the blue-glazed pottery from the Melfi Castle, of the use of a local, easily available, and low-cost raw material, such as the haüynophyre of Melfi, from which haüyne crystals were picked up to be used as a substitute for lazurite. The occurrence of haüyne is a deliberate technological choice. In fact, the use of lapis lazuli, in the coeval glass from the same context, points out that the craftsmen were familiar with the use of lapis lazuli to color vitreous materials blue, and therefore, the sources of supply and trade routes of lapis lazuli [17] must have been known and there must have been availability in the workshops of Melfi. The artisans must also have been fully aware of the difficulties of importing it and the high costs associated with its use. This may have prompted them to look for a cheaper and more readily available alternative, and the fact that Melfi Castle is built on and not far from outcrops of volcanic rocks that contain blue-colored haüyne crystals must have induced them to try it. 

All this confirms the great liveliness of Frederick II’s courts and led to the discovery of new manufacturing expedients in *that tormented world of Italian production of the first half of the XIII century* distinguished by *many phenomena of hybridism* [38].

The results obtained in this study also allow us to hypothesize that the Arab influence characterizing the ceramic and vitreous productions of Frederick’s castles and domus in Basilicata and Northern Apulia, highlighted by archaeological studies [34,39,40], could be, in this case, limited to technological procedures and style, and do not necessarily involve raw materials supply, that, in the studied ceramic, are not simply local but “zero-mile” sourced.

## Figures and Tables

**Figure 1 molecules-28-01546-f001:**
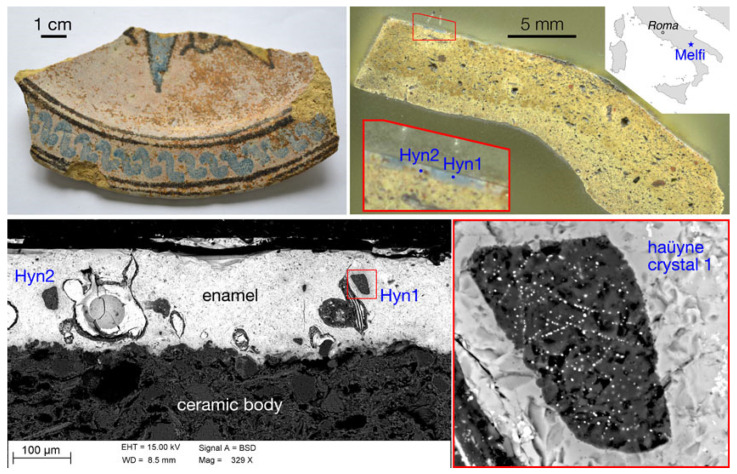
M22 sample from Melfi (upper, left); in its stratigraphic section, in the red-bordered inset, the blue zone with two blue sodalite crystals is easily recognizable (upper, right) and SEM-BSE image of the blue lead-tin glaze in which two sodalite crystals are present (bottom, left). In the red-bordered inset (bottom, right) a crystal that shows very fine opaque mineral inclusions, typical of the Vulture rocks haüyne, is present.

**Figure 2 molecules-28-01546-f002:**
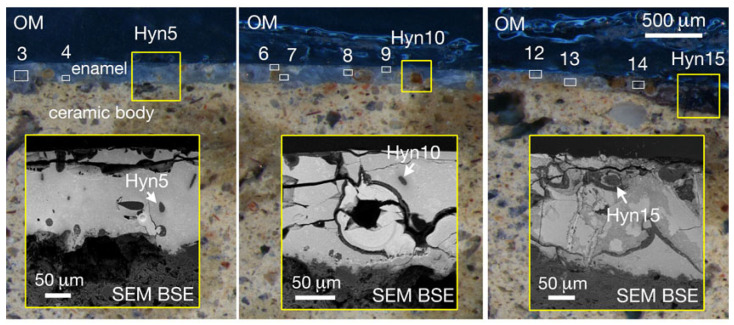
Reflected light optical microscope (OM) images of the M22 sample with the lead-tin glaze blue zone in which numerous haüyne crystals are easily recognizable; in the yellow-bordered inset SEM-BSE images of the sample are reported.

**Figure 3 molecules-28-01546-f003:**
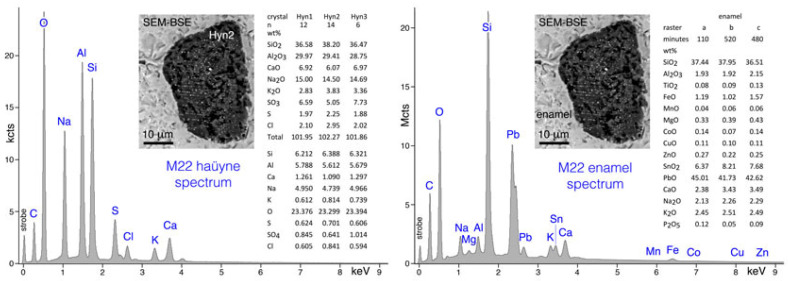
(**left**) SEM-ED spectrum of an haüyne crystal found in the lead-tin glaze of the M22 sample (data represent the compositions of different haüyne crystals); (**right**) SEM-ED spectrum of the lead-tin glaze of the M22 sample (the microanalyses, performed on raster of 320 × 60 μm with relative time acquisition, of different glaze areas are also reported).

**Figure 4 molecules-28-01546-f004:**
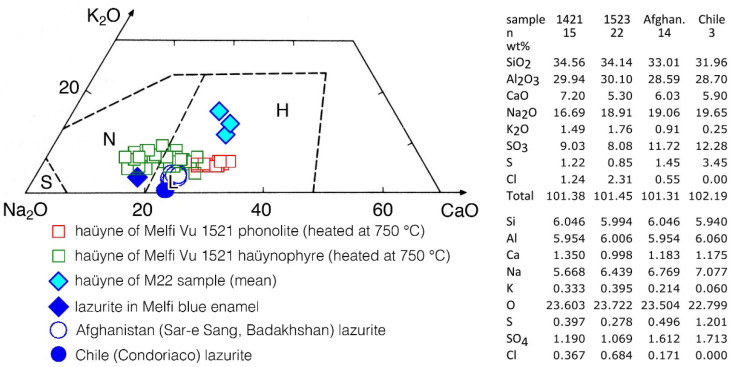
Composition of some haüyne from Monte Vulture outcrops and of the haüyne present in the lead-tin glaze of the M22 ceramic sample; lazurite of two samples (Sar-e Sang area of Afghanistan and Condoriaco area of Chile) of the Earth Science Museum of the Bari University and the mean of heated lazurite crystals of a Melfi (PZ) blue gilded and enameled glassware [1] are also plotted for comparison in the K_2_O-Na_2_O-CaO diagram. Lines separating the fields of sodalite (S), nosean (N), and haüyne-lazurite (H and L, respectively) are from Lessing and Grout [22].

**Figure 5 molecules-28-01546-f005:**
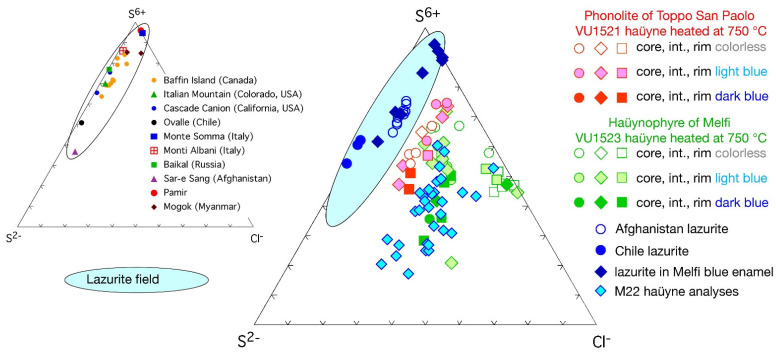
Haüyne compositions from phonolite and haüynophyre volcanics of the Monte Vulture area (Toppo San Paolo, VU1521, and Melfi, VU1523, samples) both heated at 750 °C as compared with the composition of the small crystals found in the lead-tin glaze of the M22 ceramic sample; as suggested in [23] for lazurite, the haüyne chemical data are plotted atomically with respect to S^6+^, S^2−^, and Cl^−^. Lazurite field is delimited taking into account the data reported in [1,23], in which are also plotted the analyses of two samples (Sar-e Sang outcrops of Afghanistan and Condoriaco outcrops of Chile) from the Earth Science Museum of the Bari University and the heated lazurite crystals of a Melfi (PZ) blue gilded and enameled glassware [1].

**Figure 6 molecules-28-01546-f006:**
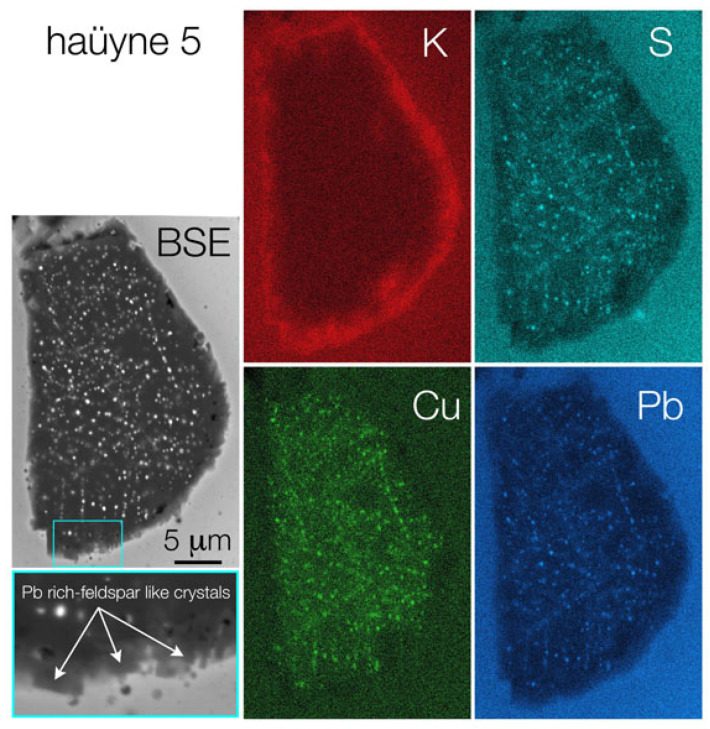
SEM BS electron images of a haüyne crystal and the relative more representative elemental X-ray maps.

**Figure 7 molecules-28-01546-f007:**
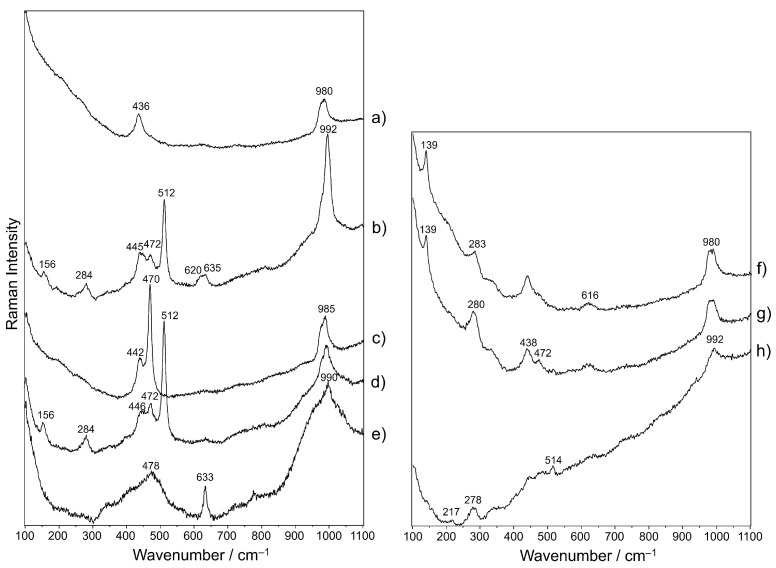
Representative Raman spectra extracted from the hyperspectra acquired on the mapped areas on haüyne crystals 3 (**a**–**e**) and 5 (**a**–**h**).

**Figure 8 molecules-28-01546-f008:**
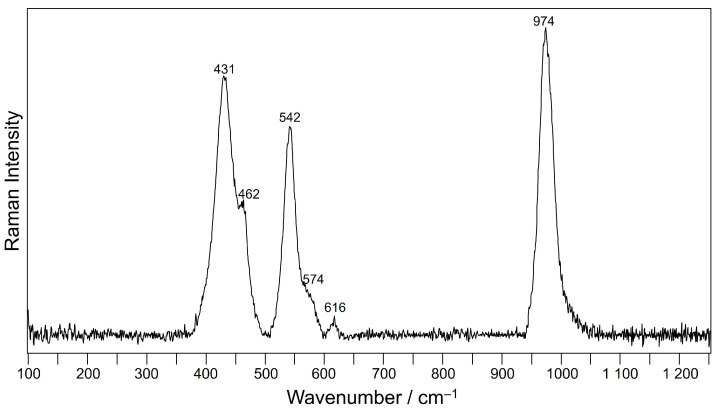
Representative baseline-subtracted Raman spectrum acquired on haüyne crystals.

## Data Availability

Not applicable.

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
