# Peer review of "A Possible Natural and Inexpensive Substitute for Lapis Lazuli in the Frederick II Era: The Finding of Haüyne in Blue Lead-Tin Glazed Pottery from Melfi Castle (Italy)"

_molecules, 2023, doi:10.3390/molecules28041546_

Round 1
Reviewer 1 Report
The content is original and the criticisms are mainly about the formatting for a clearer presentation for readers and the fact that the authors seem to be afraid to draw clear conclusions.
Abstract: formulate more clearly whether the blue pigment is imported lapis lazuli or a transformation of local hauyne or no conclusion is possible!
Lines 61-62: explain the types of objects
The intro is a bit messy. It would be clearer to first present lapis lazuli geologically and mineralogically, the constitutive phases and the 'standard' uses (frescoes, paintings, illuminated manuscripts) and the other much rarer uses of lapis lazuli as a colorant for glass or enamels , in particular the Ptolemaic glasses and ceramics, long ignored because the strong coloring power of the cobalt ions pushed in the absence of clear detection in elementary analysis of many authors not to be too surprised to have blue glasses where it does not was not possible to detect Co. The questions that the authors want/will answer should be clearly formulated.
Lines 156 to 160: two different fonts are used!
Figure 3: first peak before C: strobbe??
Figure 4: for the comparison to make sense, the Chile and Afghanistan "references" would have to have undergone a heat treatment similar to that of the Melfi samples and the enamels, with the possibility of evolution either due to heating or to the matrix silicate, which can enrich or impoverish in Na, K and Ca. This point must be better discussed.
Line 195: stress?
Lines 189-192: the similarity is not perfect: there is no overlap. Rephrase and see the remark above.
Line 206-207, can the peripheral K enrichment in Figure 5 come from the matrix glass by reaction? To be better discussed with the Melfi offset in Figure 4.
Lines 235-242: attribution of the peak at ~140 cm-1 is difficult. Traces of anatase are always possible. The formation of lead-tin yellow is also possible. The interest of figures 7 and 8 is very limited. Adding them as an appendix or supplement would make the article clearer and more effective.
Lines 285-291: In the solid phase, the diffusion coefficient of an atom depends mainly on the size of the atom and the adaptation of the host structure to this size, the latter strongly influencing the activation energy. See literature. The formation of a liquid phase changes the process. Better separate the factors and reformulate.
The conclusion is not clear: use of local materials as coloring agent or as non-coloring raw material complicating the analysis. The authors must formulate more clearly the questions, the hypotheses and list in a table the arguments for or against the use of imported lapis for glazed pottery. Authors should also discuss enamels on glass from their previous works.
Author Response
We thank the referee for a thorough review of the paper. Below are the answers to all questions.

Reviewer 2 Report
Please provide a detailed description about the ceramic assemblage at Melfi Castle and explain why this particular shard (i.e. M22) was selected for your study. How do you support your idea that the occurrence of hauyne a technological choice and, but not an accident? How many shards from Melfi castle do you think may contain hauyne in their composition?
I am not convinced that the authors have found hauyne in the shard under study. The Raman signature is not related to this mineral (see the RRUFF Raman reference spectrum of hauyne). Even the quantitative EDS microanalysis, if it is quantitative at all, do not support the idea of the occurrence of hauyne. Also, I wonder if the method adopted by the authors would end up quantitative data. The authors have not mentioned the accelerating voltage of their EDS measurements but even in low kVs there is always change of loosing light weight elements (e.g. Na) over time (considering the EDS spectra are acquired for more than 100 minute).
It is peculiar that the authors have observed covellite in the so-called hauyne crystals and have not discussed it at al. Please explain where did find covellite crystals in your SEM-EDS studies? Where did you collect your EDS data from? What is the significance of the occurrence of covellite in your data? Could the blue colour of the glaze under study was supposed be driven from the Cu-rich crystals of covellite?
Author Response

(The authors gave the same response as above.)
